# Toward a Business Resilience Framework for Startups

**Leo Aldianto** [1], **Grisna Anggadwita** [2,*], **Anggraeni Permatasari** [3], **Isti Raafaldini Mirzanti** [1] **and Ian O. Williamson** [4]

1. School of Business and Management, Bandung Institute of Technology, Bandung 40132, Indonesia; leo.aldianto@sbm-itb.ac.id (L.A.); isti@sbm-itb.ac.id (I.R.M.)
2. School of Economics and Business, Telkom University, Bandung 40257, Indonesia
3. Faculty of Business, President University, Bekasi 17550, Indonesia; anggraeni@president.ac.id
4. Wellington School of Business and Government, Victoria University of Wellington, Wellington 6011, New Zealand; ian.williamson@vuw.ac.nz
* Correspondence: grisnaanggadwita@telkomuniversity.ac.id

**Abstract:** Covid-19 has had a significant impact on the disruption of the global economic sector, including for startup businesses. This encourages entrepreneurs to carry out a continuous innovation process to become more ambidextrous and continue to innovate in an effort to futureproof their business. The paper aims to provide a business resilience framework by exploring capability (innovation ambidexterity, dynamic capability, and technology capability), behavior (agile leadership), and knowledge (knowledge stock) in startup businesses. This study uses a literature review synthesis to gain a greater understanding of startup resilience and its implementation. This study also uses a case study approach in building a framework by obtaining data from semi-structured interviews with three startups owners in Indonesia. This preliminary research has identified four propositions that will be used to develop questionnaires and data collection instruments. Thus, this study provides new insights on how startups can overcome contradictory pressures for business resilience in anticipating, dealing with, and emerging from business turbulence due to the Covid-19 pandemic by considering the factors proposed in this study. The implications and recommendations of this study are also discussed in detail.

**Keywords:** agile leadership; business resilience; Covid-19; dynamic capability; innovation ambidexterity; knowledge stock; startup; technology capability





## 1. Introduction

The Covid-19 outbreak has spread throughout the world. The World Health Organization (WHO) announced that Covid-19 is a global pandemic, causing significant economic shocks worldwide in efforts to control the virus [1]. Pandemic diseases are one of the potentially unpredictable and severe threats to the continuity of an organization's operations and infrastructure [2]. The Covid-19 pandemic has had a negative impact on all economic sectors, including SMEs, in both developed and developing countries, and startup businesses are the most vulnerable to these conditions. Startups can be described as a subgroup of SMEs—young companies aged no more than three to five years that carry out entrepreneurial activities [3]. Startups differ from large companies in terms of their organizational structure, leadership style, reactions to the environment, available resources, and the internal context in which they operate [4,5]. The global business environment has become increasingly complex due to this pandemic, so that business resilience in the SME sector becomes a determinant in business continuity.

The development of Indonesian startups is increasing rapidly, and this phenomenon is predicted to contribute to economic growth in Indonesia. However, the Covid-19 pandemic has wiped out many leading sectors in Indonesia, especially in Indonesia's digital economy. This study focuses on startup business development during the Covid-19 pandemic. Mostly,

scale-up startups have received funds from investors who demand them to continue their product and growth development. Indonesia statistics stated most companies had difficult times during the pandemic. During the pandemic, companies who were in good condition changed from 74.8% to 33%, 21.6% were in average condition, which changed to 24.6%, while startups in low (poor) condition increased from 3.6% to 42.5%, meaning that there was a significant decrease in business conditions due to the Covid-19 pandemic [6]. The situation is very dire for a startup if they intend to pivot or change to products that already exists on the market.

Both scholars and practitioners consider organizational innovation as a necessity to minimize organizational risk. Through innovation, organizations can adapt to environmental changes and reduce the impact of threats and risks [7,8]. Innovation ambidexterity is the individual's capability to balance exploration and exploitation [9]. Startups should aim to strike a balance between exploration and exploitation to improve business performance even though the growth of ambidexterity is a major challenge [10].

Dynamic capabilities and technological capabilities are needed by entrepreneurs to increase the innovation ambidexterity in an effort to improve performance and maintain business continuity. Dynamic capability refers to a company's capability to reconfigure its competence in a volatile environment [8]. Meanwhile, technological capability is the ability of individuals to acquire, disseminate, combine, and reconfigure technology resources to support and improve business strategies and work processes [11]. Another factor identified in the Covid-19 phenomenon is agile leadership, which is flexible and adaptive leadership—setting direction, setting simple and generative rules of the system, and encouraging constant feedback, adaptation and collaboration [12]. In addition, the knowledge stock can be also the basis for applying knowledge in the face of uncertain conditions, such as the global pandemic, Covid-19. Thus, knowledge stock refers to the number of elements of knowledge a company has accumulated over time.

According to [13], organizational resilience can be defined as the ability to deal with challenging conditions by ensuring the existence and prosperity of the organization. Startups are businesses that are vulnerable to survival, so startup founders may struggle in their quest for ambidexterity as startups generally suffer from limited resources and organizational structure. They have fewer resources to withstand existential-threatening crises or to secure day-to-day business [14,15]. Runyan [16] concluded several reasons why startups have been hit hardest in the face of crises, including lower levels of preparedness, higher vulnerability, reliance on local government and institutions, and greater psychological and financial impact on business owners. Organizational resilience studies are mainly carried out in large organizations and established SMEs [17–20], but empirical research about developing startup resilience at the organizational level is still limited.

This study aims to construct a conceptual model for startup resilience during the global pandemic. Therefore, it requires an in-depth analysis of what factors startups can improve to maintain startup business resilience. This study also aims to observe the innovation process carried out by a startup in managing a new adapted business model and validating their new products to sustain their business during a pandemic situation. The literature review synthesis method approach is used to build a conceptual framework that is supported by case studies on several startups in Indonesia.

This study tries to fill the gap by presenting a business resilience conceptual framework based on the context of the factors that influence innovation ambidexterity. The concept of business resilience studies at startups is still limited, so this study offers several contributions. First, this study provides insight into the impact of technological capabilities, dynamic capabilities, knowledge stock, and agility leadership in creating innovation ambidexterity and increasing the business resilience of startup. Second, this study is one of the few papers examining the innovation ambidexterity and business resilience in startups. Third, this study adds insight into business resilience factors that startups must pay attention to, as the conceptual framework built is not only used in dealing with the global

pandemic, but also in uncertain conditions. Fourth, the proposed conceptual framework can be used as a model for quantitative empirical testing in future research.

## 2. Literature Review

### 2.1. Business Resilience

Resilience is related to personality characteristics and refers to a dynamic development process [21]. According to [22], resilience is the ability to maintain the functionality of a system when disrupted or the ability to retain the elements needed to update or rearrange if an interruption changes the structure of a system's function. A resilient organization will always find ways to take chances and take advantage of situations. Donnellan et al. [23] showed that resilience is related to predicting and preventing unexpected threats. It is also important to have sensitivity, change perceptions, and manage a flexible decision-making process. Economic instability and business discontinuity require organizational agility and resilience.Linnenluecke and Griffiths [24] defined resilience as the capacity to absorb impact and recover. Meanwhile, [25] defined resilience as the ability of a system to cope with disturbances caused by external phenomena, and still remain unchanged.

According to [26], "organizational resilience is a complex blend of behaviors, perspectives, and interactions that can be developed, measured, and managed." In general, resilience is demonstrated after an event or crisis occurs [27]. Business resilience enables organizations to quickly adapt to disruptions while maintaining sustainable business operations and protecting people, assets, and overall brand equity [28]. Business resilience is "the capacity for companies to survive, adapt and grow in the face of turbulent change" [29,30]. Dahles and Susilowati [31] argue that local business and company responses to rapid changes and shocks are crucial for economic development. Resilient businesses are able to recover from disruptions and show adaptive capacity, which can cause extensive changes in the overall business concept [29]. Small businesses are very responsive to exogenous shocks because they are more flexible, adaptable and innovative than large companies [31]. Innovative and adaptive abilities play a key role in post-crisis recovery [32].

According to [33], there are three different perspectives on resilience. Scott and Laws [34] view resilience in terms of returning to the previous state, which is considered 'normality'. The second approach sees resilience as the capacity for recovery from the crisis by following the order of rescue, restoration of damaged infrastructure, and then rebuilding markets [34]. Finally, the third approach to resilience envisions a crisis to produce fundamentally different conditions. The business concept changes drastically and in an unplanned and uncontrolled way, resulting in new operating methods, new business partners and network relationships, new markets, different products, and new sources and leadership used to deal with crisis situations.

Covid-19 is a rapidly growing and unique challenge for organizations globally. Thus, businesses, especially startups, must understand the precautions that must be taken and prepare the organization to be as resilient as possible in protecting employees and maintaining operations. This includes understanding the organization's position in terms of business continuity and crisis management, specifically relating to staff, vendors, supply chains, and IT operations and infrastructure. Business resilience is responsible for identifying and understanding the main organizational and operational risks associated with the delivery of products and services, and the sustainability of operations in emergencies covering key focus areas, including: products and services, management and staff, operations and facilities, customers and vendors, and the entire value chain [28].

The concept of resilience has received little attention from systematic empirical studies [35]; the resilience-based literature has followed most theoretical approaches, focusing on conceptual development and related fields. The resilience-based literature can be broadly grouped into three general classification areas that correlate with the elements of resilience as identified by [36], which include readiness and preparedness, response and adaptation, and recovery or adjustment. Previous studies identified a number of areas for advancing resilience research, in particular, the relationship between human

and organizational resilience, and understanding the interface between organizational resilience and infrastructure. The Table 1 below shows previous studies that have focused on organizational resilience.

**Table 1.** Previous Research on Resilience.

| Authors | Topics/Concepts | | | |
| :---: | :---: | :---: | :---: | :---: |
| | **Behaviors and Dynamics** | **Capabilities** | **Strategy** | **Performance** |
| [29] | √ | | | √ |
| [37] | | √ | | √ |
| [13,18,25,38–41] | | √ | | |
| [42–47] | √ | | | |
| [48] | | | | √ |
| [49–52] | √ | √ | | |
| [28,53] | | | √ | |
| [54] | √ | | √ | |

## 2.2. Drivers of Business Resilience

*Dynamic capability*. The dynamic capability approach arises from an organization's Resource Based View [55] where organizations not only change their resources and routines, but also products and services, to be able to survive in a changing environment [8,56]. Helfat and Peteraf [57] define dynamic capability as "the capacity of organizations to intentionally create, expand, or modify their resource base" and, thus, achieve higher economic value than their competitors. In addition, dynamic capabilities are considered to transform resources into performance enhancements [58]. Teece [59] defines dynamic capabilities as "high-level competencies that determine a company's ability to integrate, build and reconfigure internal and external resources/competencies to cope with, and possibly shape, a rapidly changing business environment." Meanwhile, Zollo et al. [60] define dynamic capabilities as stable and reliable patterns of behavior that specialize in the adaptation of organizational traits towards an inclusive, sustainable, and multi-stakeholder enterprise model. Thus, dynamic capabilities are needed for business resilience and sustainability. The dynamic capability perspective has also contributed to innovation orientation [57,61]. In addition, dynamic capabilities are found in the field of marketing, a study conducted by [62] showed that in SMEs there was a positive relationship between dynamic marketing capabilities and international marketing performance in a business environment with low competition intensity and high and competitive intensity.

Dynamic capabilities include three capacities: identification and assessment of opportunities (sensing); resource mobilization to overcome opportunities and to capture the value of actions taken (seizing); and continuing to update core competencies (transforming) [63]. Sensing refers to the ability to recognize and assess opportunities and threats, which include the identification and development of technology in relation to customer needs [64]. Seizing is a critical capacity to enable organizations to act on the opportunities that have been identified. Seizing involves the following actions [65]: *designing* refers to organizational actions taken to plan and design new structures and processes; *choosing* refers to the actions an organization chooses among the various options available in terms of design and other potential solutions to seize opportunities; *acting* refer to decisions made by organizations about how to implement designs as well as decisions regarding specific choices from partners, services, processes, or business models. Meanwhile, transforming refers to managing change by reconfiguring core and complementary resources and the ability in the company's daily operations to improve them [63–69]. [63,65] broadly discusses how transforming can involve revamping routines, restructuring departments, managing specific assets, and placing governance and knowledge development structures—involving reconfiguring organizational resources.



*Technology capability*. Technological capability refers to the ability that enables companies to use and develop various technologies [70] by involving technology development, product development, production processes, manufacturing procedures, and technology estimates [71]. Meanwhile, according to [72], technological capability is the extent to which companies are good at managing information technology resources to support and improve business strategies and processes. The technological capabilities of a company include technological infrastructure, human resources consisting of technical and managerial skills, and intangible things such as knowledge assets, customer orientation, and synergy [73]. Companies can improve their business performance by utilizing their technological capabilities to increase revenue, reduce costs, or both.

Based on the results of studies from [72], technological capability consists of three dimensions, which include technology infrastructure capability, technology business development capability, and technology proactive stance. Technology infrastructure capability is a company's ability to use a shareable platform—capabilities that capture the extent to which the company is good at managing data services and architectures, network communication services, and portfolio and application services [72–76]. The technology business development capability is the ability of a company's management to imagine and exploit technological resources to support and enhance business objectives—capabilities that reflect the extent to which the company develops a clear strategic vision of technology, integrates business and technology strategic planning, and enables management's ability to understand the value of investment technology [72–74,77]. A technology proactive stance is the ability of companies to proactively look for ways to embrace technological innovation or utilize existing technology resources to create business opportunities—attitudes that measure the extent to which companies try to always be up to date with technological innovation, experimenting with new technologies, looking for new ways to increase the effectiveness of technology use, and fostering a favorable climate for trying new ways of using technology [72,76,78,79].

*Agile leadership*. Agile is a mind-set and passion for collaborating in building products, both inside and outside the team. Agile is also defined as a passion for dealing with and embracing the changes that occur during product development. Agile leaders are capable of thinking outside of the box to perfectly align an organization with its internal and external environments [80]. Agile leadership describes the ability of a leader to be quick, adaptable, and flexible in responding to unforeseen events in an unfamiliar circumstance [80]. The development of higher levels of leadership agility is important for all organizational levels, including top executives and managers [81]. Therefore, agile leadership aims to make organizations or companies more effective in collaborating and adapting to changes in building products. Agile leadership has a set of values and principles that are mutually agreed upon to make company development better—more effective as well as more enjoyable. According to [81], there are five distinct competency levels of leadership agility, such as (1) expert; (2) achiever; (3) catalyst; (4) co-creator; and (5) synergist. Understanding and living these five values in the company is difficult because the level of value is abstract and not easy to transform into practice. However, leaders need to explore the agile values and work the competencies into practices in company culture. In a conceptual framework developed by [82], consistency and agility are proposed as pillars for strategic leaders to effectively implement the core values of their business and adapt to market changes. Joiner and Josephs [81] identified four key competencies of successful agile leaders in an unstable business environment: context-setting agility, stakeholder agility, creative agility, and self-leadership agility.

*Knowledge stock*. Knowledge is a firm's essential resource in organizational learning. Acs et al. [83] stated that firms generate an abundance of knowledge, which enables entrepreneurs to identify and take advantage of opportunities. The available knowledge can be used repeatedly to develop a production process and drive innovation. Therefore, utilizing knowledge stock appropriately will lead a company's organizational learning to achieve best performance and win market competition. Papa et al. [84] found firm inno-

vation is driven by the knowledge possessed by employees. The availability of adequate and structured sources of knowledge makes it easier for startups to apply knowledge for business improvement. Knowledge stock serves as a knowledge pool used to refine existing knowledge and absorb fresh knowledge [85]. Chaudhary et al. [86] stated that the knowledge stock of a firm influences the development of its ability to acquire, assimilate and exploit external knowledge. Thus, it can support the creative process and develop various innovations. The relevance of knowledge possessed by employees is useful for innovating so as to foster internal capabilities and external opportunities [84]. Therefore, knowledge stock plays a role in streamlining processes and ways of working by utilizing all available resources from time to time to create better innovations.

Knowledge stock plays a role in increasing service or product innovation. Knowledge stock pursues dynamic market studies as well as consumer needs and depends upon its existing knowledge about customers as market knowledge [86]. According to [87], firm knowledge stock "became a significant predictor of firm innovation when existing along a high level of firm knowledge flow, as well as in firms adopting innovation strategies." Therefore, knowledge stock is a superior strategy that organizations can rely on to accelerates innovation by managing the resources, knowledge, and capabilities of the organization through experience and the latest knowledge. Knowledge stock is a provision or capital in the form of experience combined with adequate knowledge to provide ideas, values, creativity, and profit and aims to increase ideas, innovation, thinking, competence and expertise. Knowledge influences the relationship between search behavior and innovative performance [88]. Rupietta and Backes-Gellner [89] stated that knowledge creation systems consist of (types of) knowledge stock (i.e., human capital) and knowledge flows (induced by HRM practices). Thus, an organization can get savings on the cost and time required to make decisions related to innovation.

*Innovation ambidexterity*. Ambidextrous organizations have the advantage of exploiting existing competencies to enable additional innovation and explore new opportunities to drive radical innovation [90–93]. Meanwhile, Jansen et al. [94] defined ambidexterity as the ability to simultaneously pursue innovation and incremental and interrupted change. Explorative and exploitative innovation is an interdependent activity. Explorative innovation includes activities oriented to selection, improvement, and efficiency, while exploitative innovation is built on search, discovery and experimentation. Thus, exploration involves "experimenting with new alternatives" with returns that are "uncertain and far," and exploitation is "refinement and expansion of existing competencies, technologies and paradigms" with returns that are "proximate and predictable" [95]. March [95] revealed that maintaining a balance between explorative and exploitative innovation is essential for the survival of the company. In addition, Levinthal and March [96] also argued that the basic problem faced by an organization is to be involved in exploitation to ensure its sustainability today and, at the same time, carry out exploration to ensure its future sustainability. The need for the right balance between exploration and exploitation has been emphasized by [97].

### 2.3. The Impact of the Global Pandemic

The outbreak of the Covid-19 pandemic has had a major impact on global economies. The Covid-19 pandemic has impacted many sectors, including the MSME sector. According to Goldman Sachs data, 96% of SME owners in the United States stated that they had felt the impact of the Covid-19 pandemic and 75% of their businesses experienced a decline in sales. Meanwhile, as reported by online media, the General Chairperson of HIPMI JAYA, Afifuddin Suhaeli Kalla, said that the turnover of Indonesian SMEs had decreased by 70% in the past week [98].

Market demand is an important factor that determines the development and sustainability of a company. The SARS pandemic that occurred in the 2002–2004 period caused shocks and changes in consumer demand patterns [99]. Limited physical movements and decreased consumer confidence during and after the SARS outbreak led to a significant

reduction in consumption spending, leading to a highly volatile and uncertain environment for SMEs, requiring rapid adaptation to changes in the market environment associated with the labor market, supply chain, and customer demand [1]. Changes in market demand faced by various post-disaster industries have been widely studied in previous studies, where it was found that retail companies experienced changes in purchasing patterns in the post-disaster phase, such as a decrease in consumption of luxury goods [100,101] and an increase in low-cost product consumption [102].

In addition, SMEs in general have to face disrupted supply chains after a disaster, which often results in a substantial reduction in production [1]. Social restrictions cause the supply chain to also experience obstacles because various transportation routes have to be closed to reduce human movement. Recovery processes and outcomes are not only affected by the immediate impact of the disaster on SMEs, but also by long-term problems, which can include prolonged business disruption and difficulties in supplying or receiving products/raw materials [103]. The Covid-19 pandemic requires people to reduce interactions, so that several affected countries have taken steps to close the business sector for a certain period of time in an effort to reduce the spread of the virus, subsequently leading to decreases in businesses' income levels. However, to ensure business continuity for firms across disruptions [104], it is important to maintain a steady stream of income. Therefore, SMEs can take advantage of savings to prevent income disruptions and pay for daily expenses during the closed period when income is low [1,105]. Biggs et al. [106] suggest that more substantial savings have a lower risk when a disaster strikes. For companies that have no cash flow and no savings, government support during a crisis is important to ensure the survival and recovery of SMEs [105] because, apart from providing loans, the government can implement spatially targeted tax incentives to promote post-disaster revitalization, encouraging business reinvestment to help ease the pressure on the SME capital chain [107].

## 3. Research Method

This study uses a qualitative method to address conceptual research toward a business resilience framework for startups. This study focuses on startups in the acceleration phase, many of who were still facing problems with the new normal condition during the Covid-19 pandemic. For that reason, this study uses a desk review and case study approach to examine the startup's business resilience. The case study was undertaken by interviewing three startups and observing the changes that lead them to survive during the pandemic. Creswell and Poth [108] stated that the focus of the case study is the specification of cases that include individuals, cultural groups, or portraits of life. The article uses the perspective of startup owners/founders or CEOs to understand how startups adopt innovation ambidexterity and the impact of Covid-19 on business resilience.

### 3.1. Data Collection Process

This study observed and interviewed three startup companies which successfully adapted their business models during the Covid-19 pandemic. The companies have survived by changing their organizational behavior to become more agile. The data collection was undertaken between October and December 2020, through online interviews in Jakarta, Bogor, and Bandung. This study used semi-structured questions to interview respondents about their practices and experience to develop startup resilience. This strategy allows researchers to investigate the incidents and challenging issues that startups have faced during the Covid-19 pandemic. Data collection in case studies can be drawn from various sources of information because case studies involve collecting "rich" data to build an in-depth picture of a case [108].

The data collection process uses digital communication applications to facilitate recording and documentation. With limited space, the researcher seeks to understand the setting and situation, which shows the context and relies on the values that arise from the interaction between researchers, informants, and social changes. Creswell and Poth [108]

describes an analysis of a case context or setting where the case can represent itself. Therefore, this study describes the subject through a chronology of significant events, followed by a detailed perspective on some occasions.

### 3.2. Data Coding and Interpretation of Findings

This study analyzes the data using an interpretive approach from interviews and observations. This study also identifies the concept from previous literature and previous research findings related to innovation ambidexterity and business resilience. Creswell and Poth [108] revealed that case study data analysis requires multiple sources of data to determine evidence at each phase in the case's evolution. Therefore, data analysis focuses on answering research objectives.

Next, we do the grouping of similar text segments into code. We analyze the information to determine how the event occurred according to the setting. After that, we compared the coding schemes with a literature review to help understand the process. Previous studies have shown that the qualitative description approach offers subjective interpretations reinforced and supported by reference to verbatim quotations from participants [109,110]. The triangulation method is used to ensure data collection trusted by matching the interview results using independently coded triangulation of sources. Case studies require extensive verification through triangulation and member checks to help researchers check the data's validity by reviewing and comparing the data. This study takes four prepositions by identifying variables that affect innovation ambidexterity on business resilience. This study draws propositions by identifying the dynamic capabilities, technology capability and agile leadership that startups utilize during a global pandemic. The last two propositions were derived from knowledge stock and the impact of Covid-19 on startup business resilience.

### 3.3. Respondents Profile

The first respondent is a media analytic startup in Bandung, which becomes a one-stop data analytical platform. The startup provides services to help clients doing big data analytics at affordable prices. The second respondent represents a biotechnology startup in Bandung, which produces natural health products and will bring a lot of benefits for the community. The development focused on selling crude essential oil and derivative products used for specific purposes. The third respondent is a human resources startup in Jakarta, which promotes fresh graduates as a big company's talent resource. The startup builds a networking platform to link talented job seekers and companies. The platform is reliable and can meet the needs of job seekers and employers more quickly and safely. Most of the partners are universities, startup businesses, and business incubators connecting talent and job opportunities.

## 4. Findings

### 4.1. Case Study

In this study we used a case study by collecting data through interviews, archives, and observations. Informants were selected using purposive sampling techniques with criteria that include: startup owners and founders, and startups that have been running for at least two years. In addition, these startups can work well together and allow unlimited data access to researchers. A total of three startups participated in this study from several business fields including those engaged in career development platforms, health care, and data analysts. This case study aims to identify the impact of the global Covid-19 pandemic on startup resilience, and explore the factors that influence it. The cases studied are very relevant to world conditions during 2020, where all countries are affected by the Covid-19 outbreak, especially startups who must struggle to maintain their business continuity.

Observations and semi-structured interviews were carried out to identify and analyze concrete cases that have occurred in the face of the Covid-19 pandemic. Interviews were conducted online using Google Meet media with the duration of each interview between

60–120 minutes, covering several topics: the condition of startup companies during the Covid-19 pandemic, trends in product demand during the Covid-19 pandemic, and factors affecting the resilience of their businesses. We identified several variables based on the interview results for use in developing a model conceptual framework.

We used respondents' initials to keep their privacy. A1 = career development platform; B1 = data analytics; C1 = health care business. Based on the results of the interviews presented in the Table 2 below, it shows that several variables were identified including: dynamic capability, technology capability, agility leadership, knowledge stock, innovation ambidexterity, and business resilience.

Based on the interviews, we found that startups have applied their dynamic capabilities to adapt the business to the situation by identifying customer needs, conducting market research, and setting priorities. Startups also seize business by taking action based on the opportunities they have identified. The transformation is carried out by startups with changes in work patterns by implementing strategies that are tailored to the conditions of the organization. In addition, it was found that startup companies were able to provide physical infrastructure to support work processes, and facilitate online platforms for employees who work at home. In addition, startups were also trying to develop their business in a pandemic situation by understanding the value of investing in technology without cutting the main budget. Meanwhile, they also had a proactive technology stance by taking the initiative to provide a technology platform so that people could access their products. Thus, we conclude that startups have the technology capabilities to make their businesses more resilient, with innovative ways to explore and exploit their technological capabilities.

The results of the interviews show that startup leaders have the agility to make organizations more flexible and adaptive; they also seek to collaborate with other parties to future proof their businesses in dealing with pandemic situations. We also found that respondents of this study stated that they have an adequate educational background and experience, so they have the knowledge stock to innovate and make their businesses resilient in the face of a pandemic situation. The orientation to innovate by exploring through the process of improvement and efficiency was conveyed by respondents during the interview process. The company increases its innovation by taking advantage of its new innovation-oriented business.

**Table 2.** Interview Results.

| Variables Identified | A1 | B1 | C1 | Conclusion |
|---|---|---|---|---|
| **Dynamic Capability** | | | | |
| Sensing [64] | The first thing we did was customer discovery. | The challenge we're currently facing is choosing our priorities | We need to conduct some research on market industry needs, because they are the one that will pay us, so we need to know their demands first. | Startups identify customer needs by conducting market research and setting priorities |
| Seizing [65] | Data . . . what distinct us from the rest is how we conduct user acquisition. Other companies conduct user acquisition through social medias, SIO, etc, focusing on the user or the jobseeker. In our case, we focus on universities. | . . . gain more market and market penetration | On social distancing we also have an increase on disinfectants, because people were looking for it, and we need to send our products on huge volume | Startups take action based on the opportunities they have identified |
| Transforming [63,65] | The strategy to increase sales is: first, to finish every aspect of product legality. Second, how we shift to doing workshops intensively. | Specifically for the employees, there's an additional change in their KPI. The KPI used to be performed in every 2 weeks, now it's performed daily, and it needs to be performed by each employee independently. | We're intensifying on reseller aspect. Before the pandemic, we chose not to be focused on partnerships or consignment, so we've never really focused on reseller. | Startups make changes to work patterns by implementing strategies that are tailored to the conditions of their organization |

**Table 2.** *Cont.*

| Variables Identified | A1 | B1 | C1 | Conclusion |
|---|---|---|---|---|
| **Technology Capability** | | | | |
| Technology infrastructure capability [72] | Regarding the infrastructure, sometimes the workers complained about it. So additional investment is needed. | Including now, since we rent 2 office buildings with capacity of 60–70 people, which is currently resided by 15 people. We might not renew the rent next year, since we are getting used to WFH. | WFH is not completely work from home. During production time, they need to do it in the office/factory. | Startups provide physical infrastructure in the form of office buildings to support the work process, and facilitate online platforms for employees who work at home |
| Technology Business Development Capability [72] | We don't really have any problem during this pandemic. The ads are still going, no budget is being cut. We only cut the one that didn't perform. | We're focusing on our tech team to work hard, so when the system is developed and ready, we're going to offer it to companies that need it. | We promote our product digitally. | Startups have the ability to understand the value of investing in technology without cutting into the main budget. In addition, startups are also preparing systems for developing their businesses in the face of a pandemic by maximizing the development of digital technology. |
| Technology Proactive Stance [72] | Some improvements need to be made, so we're focusing on digital improvement. | We help by providing a platform, where people managing the pre-work card system can utilize the automation, for example, they can send the certificate to every single participant by email in just one click. | We were finally able to do online workshop. We showed the audience some demonstration regarding our product and sent them the materials so they would be able to use the app at home. | Startups take the initiative to provide technology platforms so that people can access their products. |
| **Agility Leadership** | | | | |
| [12,80,111] | We define this organization more flexibly and adaptively, for example with a work from home system. Workers continue to carry out their responsibilities by working at home. We're focusing on collaborations. We also have a lot of students, close to 100.000. And other start-ups feel that collaborations will be very beneficial. | We emphasize to the teams that the success of the organization is our responsible, so we have to the same purpose to achieve that. ..we realized that developing product by ourselves requires some more research to be conducted, so we finally decided to collaborate | We have changed our business process soon since this pandemic with include the crisis management. Collaborations are something we can do during this pandemic, and the cost is really affordable too. | Startups use organizations more flexibly and adaptively. In addition, startups also take the initiative to collaborate in maintaining their business. |
| **Knowledge Stock** | | | | |
| [84,85,112] | Our educational background provides an essential knowledge base of how to run a business. | We have several years of experience in this business, and previously we had experience in other fields even though the business was not running, but at least it gave us experience on how to maintain business. | We try to provide knowledge about health to the public through online workshops while promoting our products. Thus, we do not only sell the products, but also provide knowledge to the community. | Startups have a knowledge stock with an adequate educational background and experience. They use their knowledge stock to resilience their business. |
| **Innovation Ambidexterity** | | | | |
| Exploration [10,95] | We can try to do things differently in the future, like tackling different approaches on some of our usual activities. | We have created some new innovations during Covid-19. One of them is we have established product for map-based analytics. | We realized that continuous innovation is very important. So we do that, we try to innovate in product development. | Startups make innovations with an orientation towards improvement and efficiency. |
| Exploitation [10,95] | Presently, our tech teams such as programmer, etc. are focused on developing our existing technology in order to extend our revenue stream or maybe create a new business line. | We created the application in hope of restoring balance between health and economy. We facilitate the big data technology and the mobile app. | We have some products in development this year; one of those is hand sanitizer. At first, we only have disinfectant, and we are finally able to develop hand sanitizer product during this pandemic. | Startups seek to exploit with a priority to produce new innovations |

<div align="center">**Table 2.** *Cont.*</div>

| Variables Identified | A1 | B1 | C1 | Conclusion |
|---|---|---|---|---|
| | | **Business Resilience** | | |
| [22,113] | We have 40 people in total in our team; we managed to keep everything in check this far. But if there comes the need to layoff, it wouldn't be just because of Covid-19, but it because their poor performance and not being able to adapt. | Efficiency policy and investment or innovation policy. For efficiency policy, the first thing we did is pay cut. Then, we applied Work From Home (WFH) policy. The other policy is for innovating or invests. | Since 50% revenue come from bazaar, the revenue stream is highly affected. That's why our strategy is to create online workshops and build networks | Startups maintain system functionality under compromised conditions. They try to take precautions by designing business strategies to deal with the pandemic. |

### 4.2. Conceptual Framework Development

The conceptual model framework in this paper is built based on the phenomena that occur and is supported by existing literature. Dynamic capabilities and ambidexterity enhance a company's ability to adapt to an uncertain and dynamic business environment because this ability can create critical knowledge for innovation through organizational learning [7,95]. Organizations with higher levels of resilience pursue dynamic capability and ambidextrous strategies [15]. Therefore, Proposition 1 is the following:

**Proposition 1.** *Dynamic capability has a significant influence on startup business resilience mediating by innovation ambidexterity.*

Previous literature shows that achieving innovation ambidexterity in SMEs depends on key resources and capabilities, such as information technology resources [114]. Technological capabilities can increase the exploitation of the ability to take advantage of existing market opportunities and explore new opportunities to meet the challenges of emerging markets. Technology capabilities can help exploit organizational innovations that are developed and accumulated from previous experience [115]. Therefore, higher technological capabilities will direct organizations to seek more knowledge and utilize current resources [115] thereby strengthening organizational learning, which will facilitate exploitative innovations [96,116]. Based on the results of the study by Wiratmaja et al. [117], technology capability has a positive effect on the innovation ambidexterity and has an indirect effect on company performance. Technology capability is an important facilitator of organizational exploitation as, one of the organizational capabilities in realizing business resilience; it can also foster exploratory innovation through increased use of organizational technological resources [118]. Proposition 2 of this study is the following:

**Proposition 2.** *Technological capability has a significant influence on startup business resilience mediating by innovation ambidexterity.*

Agile leadership is how the leader explores its values and principles, in daily development. Therefore, applying agile leadership is a challenge, mostly because the organization has difficulties accepting environmental changes into their initial development plans. In addition, agile leadership focuses on interaction and collaboration between units and stakeholders to be responsive to changes. The organizational leader needs to be aware of these powerful trends and develop agile companies–organizations, that anticipate and respond to rapidly changing conditions in ways that effectively manage both technical and stakeholder complexity [81]. Therefore, corporate leaders need to embrace the values and principles of agile through exploration. Proposition 3 of this study is the following:

**Proposition 3.** *Agile leadership has a significant influence on startup business resilience mediating by innovation ambidexterity.*

Lee and Huang [85] argued that "the existing knowledge stock will drive innovation and the way firms manage the tension of knowledge exploration and exploitation". This approach is a method applied by developing knowledge as a whole as a measurement of knowledge management. Wu and Shanley [88] stated that "the characteristics of knowledge stock moderate the effect of exploration on innovation, building knowledge reservoirs more suitable for facilitating knowledge application and utilization, and vice versa." Meanwhile, Chaudhary [86] inquired the role of existing company knowledge by proposing the role of the breadth and depth of existing knowledge in developing the company's absorption capacity. This helps each member to identify all the benefits for a company. It also provides space for members to conduct discussions and debates about internal management because "knowledge stock enhances the performance of exploitative learning and determines the direction and effectiveness of exploratory learning" [85]. This supports Acs et al. [83] who argued "knowledge spillovers come from the stock of knowledge, and there is a strong relationship between such spillovers and entrepreneurial activity." Thus, they can measure the impact of a decision in providing the best solution based on corporate culture. Therefore, in the face of dynamic changes and business competition, knowledge stock is the key to innovation ambidexterity for organizational success. The next propositions of this study are the following:

**Proposition 4a.** *Knowledge stock has a significant influence on startup business resilience mediating by innovation ambidexterity.*

**Proposition 4b.** *Knowledge stock has a significance influence on business resilience.*

Existing literature on innovation ambidexterity has focused primarily on large and multi-unit companies [119–121]. However, researchers acknowledge that empirical findings in large companies cannot be generalized to small companies. SMEs face more problems in achieving ambidexterity, because they have limited managerial expertise, unstructured procedures, and less formal systems for coordinating antithetical activities. Previous studies have found evidence that SMEs tend to achieve different innovations compared to large companies [9]. In addition, previous research has provided sufficient arguments that emphasize the difficulty of small companies in achieving innovation ambidexterity, but there are few studies that analyze this phenomenon in the specific context of SMEs [9,10,122]. Thus, this study attempts to explore innovation ambidexterity of startup businesses in facing the global pandemic (Covid-19) in which their capability to balance the explorative and exploitative enables their businesses to survive the global economic shock. Therefore, Proposition 5 of this study is the following:

**Proposition 5.** *Innovation ambidexterity has a significant influence on business resilience.*

The Covid-19 pandemic is a disaster that has had a very serious impact not only on public health, but also on national economies. This study adopts a survey of the impact of Covid-19 on the creative industry conducted by the Indonesian Alliance Foundation and the ITB School of Business and Management [123]. The Pandemic Impact Survey on Creative Industries in Indonesia was conducted to find how the pandemic impacted various sectors, including the efforts made by creative economy actors to be able to survive and develop in a pandemic situation. This study identifies that the impact of Covid-19 affects the innovation ambidexterity and knowledge stock owned by startups in increasing business resilience. This shows that startups can maintain their business by managing ambidexterity innovation capabilities and knowledge stock. This study proposes a conceptual model in an effort to seek new knowledge and new business patterns in the midst of a pandemic by startups to be able to survive and rise to seize the various opportunities that arise in this condition. The impact of the Covid-19 pandemic in this study consists of business challenges, competition strategy, value chain, and source of funding [123]. The propositions of this study are following:

**Proposition 6a.** *Innovation ambidexterity has a significant effect on business resilience moderates by the impact of global pandemic moderates.*

**Proposition 6b.** *Knowledge stock has a significant effect on business resilience moderates by the impact of global pandemic moderates.*

Figure 1 illustrates the proposed conceptual model of business resilience for startups.

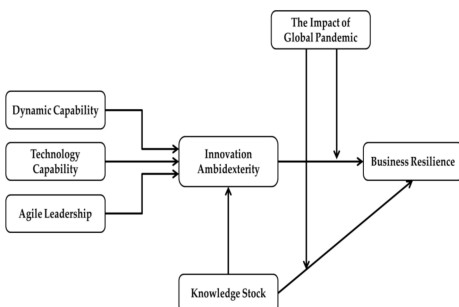

**Figure 1.** The proposed Conceptual Model of Business Resilience.

## 5. Discussions and Conclusions

This model presents the concept of business resilience in the face of the global Covid-19 pandemic. Based on a synthesis of the literature review, there is a strong focus on building theories and definitions of organizational resilience. However, the literature lacks evidence proving the theory empirically and exploring the dynamics underlying resilience [2]. As a result, there is a lack of critical discussion about how systems, such as organizations, can achieve higher resilience levels. Thus, further exploration and research need to be carried out, primarily focusing on applying empirical methods, such as case studies and surveys, that can significantly add and validate theoretical constructs. Although several case studies are found in the literature, only a few have focused on organizational resilience [2]. This research presents a growing insight into organizational resilience, especially concerning startups. Factors that influence business resilience's success include innovation ambidexterity, dynamic capability, technology capability, agility leadership, and knowledge stock. The impact of global pandemic (Covid-19) is a factor that can strengthen or weaken the effect of innovation ambidexterity and knowledge stock on business resilience.

For theoretical implications, this study responds to the phenomena during the period 2020, regarding the impact of the spread of Covid-19 on all economic sectors by conducting an extensive literature review on important variables. This study extends the discussion related to dynamic capability, innovation ambidexterity, and business resilience. These findings then concluded several relationships between relevant variables. This study proposes several propositions based on a synthesis from the literature review and case study, to be tested using primary data obtained from questionnaires distributed to startups.

A conceptual model has been developed to answer research questions by investigating the relationship between six propositions. Four propositions developed related to the relationship between innovation ambidexterity, dynamic capability, technology capability, agile leadership, and knowledge stock. Based on interviews, all startups agreed that they could find new business opportunities to increase revenue streams. Dynamic and Technology capability helps companies to be able to carry out business resilience quite well. The finding supported previous research by [15,115]. Study results from [124] also propose a dynamic capability model, where innovation outcomes are one of the consequences of dynamic capabilities. Based on the results of the interviews, it shows that the dynamic capabilities of startups are seen as a response to changing market needs [17,18,62]. This capability can be seen from the emergence of new products/services, more efficient processes, or other changes that aim to respond to changes in environmental conditions during the Covid-19

pandemic [12,46,62]. Startups are also preparing systems to develop their businesses in the face of a pandemic by maximizing the development of digital technology. The results of the study by [125] revealed that quarantine and isolation during the Covid-19 pandemic caused cognitive dissonance or adaptability in India. However, the working from home method provides an opportunity for a person to achieve a work-life balance and looks forward to an innovative path for working from home by maximizing their technological capabilities. Moreover, the startups indicate that agile leadership is needed to assist management in analyzing and making decisions regarding innovation processes, supported by [81].

Meanwhile, the knowledge stock variable has a strategic position in increasing innovation ambidexterity and business resilience, as stated by [88]. However, all startups still argue that knowledge management is still to be developed in the internal company [85]. That is why they can survive by utilizing knowledge stock as a strategic resource in facing the global pandemic. It concludes that all four variables affect the innovation ambidexterity of startup companies. The validity and reliability of each influencing factor will also increase business resilience in startups.

At the same time, the Covid-19 impact variable is closely related to the progress of startup 3, where they have many problems adapting to the global pandemic, mainly due to changes in company behavior and reduced demand for talent. The unstable economic situation caused large companies to close recruitment. Companies have adapted by making use of the available resources or reducing their employees. The three startups also experienced the cause of the declining demand; it's just that the value of the business model owned by startups 1 and 2 can adapt more quickly to pandemic conditions. Therefore, proposition six showed that the global pandemic's impact moderates the relationship between innovation ambidexterity and business resilience. This statement also extends previous research by [123].

For practical implications, this study focuses on startups, where startup businesses are the most vulnerable businesses in facing the Covid-19 pandemic. This statement is supported based on the results of a case study conducted by interviewing three startups. From the results, we argue that, for startups to successfully maintain their business in the face of the global pandemic (Covid-19), they must consider knowledge (knowledge stock), behavior (agility leadership), and capabilities (innovation ambidexterity, technology capability, and dynamic capability). The health, economic and human crises amid the Covid-19 pandemic have caused changes in every dimension of their social order [125]. These factors are critical for startup businesses to create business resilience in the midst of economic and environmental shocks caused by the global pandemic (Covid-19).

The model leads startups to adapt more quickly, create innovations, renew business models, and build effective strategies. The model developed for startup owners can be useful for anticipating change and reducing the risk of uncertainty in business resilience during a global pandemic. Agile leadership and knowledge management have an important role to play in encouraging innovation in situations of uncertainty. Therefore, in this case, startups will be better prepared to make breakthrough innovations following the current market needs. In addition, the impact of the global pandemic also moderates the effect of innovation ambidexterity and knowledge stock on business resilience.

## 6. Recommendation and Future Research Direction

This study has several limitations, one of which is the lack of longitudinal studies [126]. Thus, this study will serve as the basis for conducting longitudinal studies for further research to investigate these factors; and will try to test the research hypotheses using structured instruments. In addition, a limitation of the large amount of literature reviewed is that research generally only explores the time period when the phenomenon occurs. However, this study assumes that the current global Covid-19 pandemic has provided startups with unpreparedness, so that the proposed model is expected to be a reference in considering factors when facing other global pandemics. Business resilience is the ability of a business to survive in a critical condition, so that business resilience in this study can

be widely assumed not only in dealing with the Covid-19 pandemic, but also in dealing with all critical situations. Thus, further research is needed to present significant research.

The recommendation for further research is to develop a simulation model to represent the actual startup conditions. This conceptual model will provide the basis for further empirical research that investigates the influence of the factors identified in this study on business resilience in startups. This proposed conceptual model can be used by startups to increase business resilience based on their needs. In addition, the factors proposed, which include innovation ambidexterity, dynamic capability, technology capability, agility leadership, and knowledge stock, can be considered in an effort to increase business resilience by taking into account the impact of the global pandemic described in this study.

**Author Contributions:** Conceptualization, L.A., G.A., and A.P.; methodology, G.A.; data collecting, L.A., G.A., A.P., and I.R.M.; data processing, G.A., and A.P.; analysis, L.A., and I.O.W.; writing—original draft preparation, G.A., and A.P.; writing—review and editing, L.A. All authors have read and agreed to the published version of the manuscript.

**Funding:** This research was funded by the School of Business and Management, Bandung Institute of Technology, Indonesia.

**Institutional Review Board Statement:** Not applicable.

**Informed Consent Statement:** Not applicable.

**Conflicts of Interest:** The authors declare no conflict of interest.

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
