# Peer review of "Toward a Business Resilience Framework for Startups"

_sustainability, doi:10.3390/su13063132_

Round 1

Reviewer 1 Report

The theme is appropriate for the journal, resilience being the way to overcome adversity.

The abstract is correctly structured. You must advance the number of interviews conducted.

A desk review on business resilience should be done after the introduction.

The first part of the methodology (2.1 Business Resilience) is precisely a theoretical review, it should not go into methodology. Much of the methodology is a theoretical review. The work must be structured correctly, indicating in methodology the techniques used to collect the data and analyze that data.

Point 3 begins by explaining the case study. It would be part of the methodology. A section of results must also be specified.

The discussion should refer to other studies and verify whether the results obtained in the current one reaffirm what has already been said or are novel.

The work needs to be restructured with a scientific investigation style, clearly setting out the objectives, theoretical review, methodology and results.

Author Response

Dear Reviewer,

Thank you for the review which is very useful in improving the quality of this article. We have made revisions according to your input. Please find in the attachment.

Reviewer 2 Report

The impact of COVID-19 for all aspects of human life, including business, cannot be ignored and seems to be long-lasting. This is the point of departure for the authors, which is an important point. 

Another important aspect in the literature is the dynamic capability approach. The interest in DCs has grown significantly in the last two decades, and the concept seems to be interesting not only for scholars but also for managers. Therefore, the approach by the authors focuses on an important achievement of the business discipline.

Keeping these aspects in mind, I would like to point out to the following:

First of all, I am not sure how the literature was selected especially with respect to Table 1. Did the authors use any criteria?

Second, I did not see any methodological overview of how the case study was conducted in the materials and methods section. Authors start the case study section (section 3) by focusing on the criteria, but I would suggest a more detailed approach on the methodology section. 

Third, the table with startups' responses seems to remain uncommented and self-contained. This should be explored further in section 3.

The weakest part of the paper is section 5, which should actually be the strongest one - provided the huge stock of papers on dynamic capabilities, resilience, ambidexterity of innovation and so on... For this section, I would suggest to take a look at the following papers in order to clearly see how the case study of the authors can be situated in the ongoing framework of the related concepts, though similar papers can also be found in related journals:

https://doi.org/10.1002/gsj.1122

https://doi.org/10.5465/annals.2016.0014

https://doi.org/10.3389/fpsyg.2020.575491

 https://doi.org/10.3390/su13020579 

I would especially expect the authors to take a look at Sustainability and how previous scholars used these concepts in tailor-made case studies and conceptual contributions.

Good luck!

Author Response

(The authors gave the same response as above.)

Round 2

Reviewer 1 Report

All requested changes have been made

Reviewer 2 Report

Dear authors,

thank you very much for the much improved version of the article which will crate a visible impact both to research and to practice.